# A new family of StART domain proteins at membrane contact sites has a role in ER-PM sterol transport

Alberto T Gatta[1†], Louise H Wong[1†], Yves Y Sere[2†], Diana M Calderón-Noreña[2], Shamshad Cockcroft[3], Anant K Menon[2], Tim P Levine[1*]

[1]Department of Cell Biology, UCL Institute of Ophthalmology, London, United Kingdom; [2]Department of Biochemistry, Weill Cornell Medical College, New York, United States; [3]Department of Neuroscience, Physiology and Pharmacology, University College London, London, United Kingdom

**Abstract** Sterol traffic between the endoplasmic reticulum (ER) and plasma membrane (PM) is a fundamental cellular process that occurs by a poorly understood non-vesicular mechanism. We identified a novel, evolutionarily diverse family of ER membrane proteins with StART-like lipid transfer domains and studied them in yeast. StART-like domains from Ysp2p and its paralog Lam4p specifically bind sterols, and Ysp2p, Lam4p and their homologs Ysp1p and Sip3p target punctate ER-PM contact sites distinct from those occupied by known ER-PM tethers. The activity of Ysp2p, reflected in amphotericin-sensitivity assays, requires its second StART-like domain to be positioned so that it can reach across ER-PM contacts. Absence of Ysp2p, Ysp1p or Sip3p reduces the rate at which exogenously supplied sterols traffic from the PM to the ER. Our data suggest that these StART-like proteins act *in trans* to mediate a step in sterol exchange between the PM and ER.

**\*For correspondence:** tim. levine@ucl.ac.uk

[†]These authors contributed equally to this work

**Competing interests:** The authors declare that no competing interests exist.

## Introduction

Although lipids are synthesized only in specific locations in the cell and must be exported to populate membrane-bound organelles, the mechanisms of intracellular lipid traffic are still uncertain. Each organelle has a unique set of lipids, which in some cases (for example, mitochondria) cannot be delivered by vesicles. Even for organelles linked by the secretory pathway, non-vesicular mechanisms dominate for both phospholipids and sterol (*Holthuis and Levine, 2005*; *Voelker, 2009*; *Holthuis and Menon, 2014*). Sterols are critically important lipids that are synthesized in the endoplasmic reticulum (ER) and trafficked mainly to the plasma membrane (PM). Traffic is very fast (t½ < 5 min), bidirectional, and independent of the secretory pathway (*Pagano, 1990*; *Simons and Ikonen, 2000*; *Baumann et al., 2005*; *Mesmin et al., 2011*), so non-vesicular mechanisms must exist to transfer sterol across the cytoplasm between the ER and PM.

Non-vesicular sterol transport is likely to be mediated by sterol-specific lipid transfer proteins (LTPs) with the ability to extract sterols from membranes, effectively solubilizing them for transport through the cytoplasm. Models vary in the envisaged distance across which LTP-sterol complexes must diffuse. When the entire ER is considered to exchange lipid with the entire PM, the presumed diffusion distance is up to 50% of the cell diameter. Other models specify that LTPs only diffuse between ER and PM where they form membrane contact sites (MCSs) with interorganelle gaps ∼30 nm (*Helle et al., 2013*; *Prinz, 2014*). The ER forms MCSs with many organelles, and MCSs have become a significant subject of research as MCS-specific proteins have been identified. The dominant classes of MCS proteins are tethers (*Pan et al., 2000*; *de Brito and Scorrano, 2008*; *Lahiri et al., 2014*), regulators of calcium traffic (*Takeshima et al., 2000*; *Wu et al., 2006*), lipid biosynthetic enzymes

**eLife digest** Membranes are crucial structures for cells that are made primarily of fat molecules. The most important membrane is the external one that surrounds cells and keeps the outside world out and cellular contents in. The single most common fat component in the external membrane is cholesterol, which makes the membrane rigid and better able to withstand the outside world. So even though excess cholesterol contributes to diseases such as heart disease, stroke and Alzheimer's, the external membrane of every cell needs about a billion cholesterol molecules for its normal function. But how do cells manage the traffic of these molecules to their destination?

It is known that when external membranes are short of cholesterol they make it at a different cellular location. There is an internal network—called the endoplasmic reticulum—that spreads just about everywhere throughout the cell. This network is where fats like cholesterol are made when the cell has not got enough, and where they are converted into an inert form when the cell has too much. What is not known is how cholesterol moves to and fro between this network and the external membrane.

One theory is that cholesterol and other fats move only where the internal network comes into close contact with the external membrane, without quite touching. This theory comes in part from the finding that many of the proteins found in the narrow gaps between the internal network and the external membrane are capable of transferring fats across the gap. However, one of the missing supports for this theory is that no protein that transfers cholesterol across this gap has been found.

Gatta, Wong, Sere et al. used computational tools to scan the database of known proteins for those that might be able to transfer cholesterol, and found a new family of fat transfer proteins. Further experiments showed that these proteins only bind to cholesterol out of all the fats. Next, Gatta, Wong, Sere et al. studied what the proteins do in cells, but instead of looking at the proteins in human cells they studied the related proteins in yeast. This is because the details of both the traffic of cholesterol and contacts between the internal network and the external membrane are in many respects understood better in yeast than in human cells.

Gatta, Wong, Sere et al. found the cholesterol transfer proteins were embedded in regions where the internal network was in close contact with the external membrane. Also, in cells that lacked these proteins, cholesterol added to the external membrane had difficulty transferring to the internal network.

These results together suggest that the newly identified lipid transfer proteins exchange lipids between the plasma membrane and endoplasmic reticulum at membrane contact sites. Further research is required to understand in detail how these proteins work.

(*Vance, 1990*; *Pichler et al., 2001*; *Tavassoli et al., 2013*), and LTPs including ceramide transfer protein (CERT) and oxysterol binding protein (OSBP) homologs (*Vihtelic et al., 1993*; *Levine and Munro, 2001*; *Hanada et al., 2003*; *Peretti et al., 2008*; *Rocha et al., 2009*; *Toulmay and Prinz, 2012*; *Maeda et al., 2013*; *Mesmin et al., 2013*). Many of these LTPs contain a linear motif (two phenylalanines in an acidic tract = 'FFAT') that binds to VAP, a conserved integral ER membrane protein (*Loewen et al., 2003*), and these LTPs target an MCS by containing both a FFAT motif for ER targeting and a second targeting domain for the other organelle.

Finding so many LTPs at MCSs has led to the widespread expectation that lipid transfer occurs there (*Holthuis and Levine, 2005*; *Prinz, 2010*; *Vance, 2015*), but there is little direct evidence for this. Some of the best evidence relates to CERT and OSBP, which bind ceramide and sterol respectively. Both CERT and OSBP have FFAT motifs, and their recruitment causes VAP to redistribute from the whole ER into MCSs, indicating that CERT or OSBP is physically bridging the MCS (*Mesmin et al., 2013*; *Kumagai et al., 2014*). While CERT mediates non-vesicular ceramide traffic, there is some doubt that OSBP homologs mediate all sterol traffic because deleting Osh4, the major OSBP in yeast has no effect on ER → PM or PM → ER sterol transport (*Raychaudhuri et al., 2006*) (Sullivan and Menon, unpublished). Also, deleting all seven OSBP homologs in yeast only reduces PM → ER sterol traffic by ~twofold (*Georgiev et al., 2011*), suggesting that other mechanisms exist. Apart from OSBPs, the other sterol specific LTPs are members of the Steroidogenic Acute Regulatory Transfer (StART) proteins. Among 15 human StARTs, the founding member of the family (StARD1) transports cholesterol into

mitochondria in steroidogenic cells, and StARD4 transports cholesterol from the late secretory pathway to the ER (*Mesmin et al., 2011*). Budding yeast has no canonical StART proteins, but has Coq10p and Ups1-3p, which are distantly related StART-like proteins that bind non-sterol lipids (*Barros et al., 2005*; *Connerth et al., 2012*).

Here, we identified a large new protein family distantly related to StART proteins. We found that Ysp2p, one of six StART-like proteins in yeast, has two StART-like domains both of which bind sterol. Along with three other yeast StART-like proteins Ysp1p Sip3p and Lam4p, it is anchored in the ER at contact sites with the PM. In addition, its function requires anchoring to these sites in such a way that its StART-like domain can reach out across the contacts. Finally, loss of Ysp2p, Ysp1p or Sip3p reduces the rate of transfer of sterol from PM to ER, consistent with these proteins mediating a component of sterol transport.

## Results

### A family of membrane anchored StART-like proteins includes Ysp1p, Ysp2p and Sip3p

To identify novel sterol transfer proteins, we used StART domains to seed the homology tool HHpred (*Soding et al., 2005*). This has been successfully used in structural alignments to identify remote homologs for other LTPs, including TULIPs in tricalbins, and PRELI domains in Ups1-3p (*Kopec et al., 2010*; *Connerth et al., 2012*; *Schauder et al., 2014*).

We found a large family of eukaryotic proteins containing StART-like domains (*Figure 1* and *Figure 1—figure supplement 1*), which are distantly related to other domains in the StART superfamily, such as MLN64, CERT, Coq10p and Bet-v1 (*Figure 1—figure supplement 2A*). In terms of sequence alone there are few conserved residues (*Figure 1B*), so alignment requires inclusion of predicted secondary structure (*Figure 1—figure supplement 2B*). The StART-like domain is present in three human proteins (GramD1a-c), and six proteins in budding yeast (Ysp1p, Ysp2p, Sip3p, Lam4p, Lam5p and Lam6p). Because *Saccharomyces cerevisiae* duplicated its genome ~10 million years ago, related fungi have just three family members, one each for the pairs of paralogs Ysp1p/Sip3p, Ysp2p/Lam4p and Lam5p/Lam6p (*Figure 1A*). The StART-like domains in Ysp1p and Sip3p are divergent compared to those of Ysp2p, Lam4–6p and GramD1a-c (*Figure 1—figure supplement 1*).

Importantly, most proteins in the wider family combine the StART-like domain with different accessory domains that mediate interactions with membranes, particularly GRAM domains in the pleckstrin-homology (PH) superfamily and predicted transmembrane domains (TMDs) (*Figure 1A* and *Figure 1—figure supplement 2C*). The presence of a TMD is a key observation for a proposed LTP, because the TMD will anchor the protein to one membrane, so if the LTP is to traffic a lipid to another compartment, it must act at an MCS where the gap can be bridged by a single protein or protein complex (*Olkkonen and Levine, 2004*).

### StART-like domains in Ysp2p and Lam4p all solubilize sterol

The overriding property of any StART-like domain is specific binding to a lipid or other hydrophobic ligand. To determine if the regions we identified as StART-like domains bind lipid, we expressed the predicted yeast and human domains in bacteria. The only StART-like domains that we could express as soluble proteins in bacteria were the four StART-like domains of Ysp2p and Lam4p (*Figure 1A*), the most soluble being the second domain of Lam4p (called Lam4S2), so we tested if Lam4S2 binds eukaryotic lipids. We incubated purified protein with permeabilized human cells in which all lipids had been radiolabelled with [$^{14}$C]-acetate. Re-purified protein contained a single labelled lipid that co-migrated with cholesterol by TLC, but no phospholipids were co-purified (*Figure 2A* and *Figure 2—figure supplement 1A*). Sterol binding by Lam4S2 during re-purification indicates a high affinity interaction that solubilizes the hydrophobic lipid, similar to known StART domains.

We next quantitatively studied the sterol binding properties of StART-like domains using the fluorescent sterol dehydroergosterol (DHE). DHE closely mimics ergosterol, the major yeast sterol (*Georgiev et al., 2011*; *Maxfield and Wustner, 2012*) and acts as FRET acceptor for tryptophan with a Förster radius of 1.6 nm (*Loura et al., 2010*). One of the tryptophans in Lam4S2 and related sequences is in the predicted binding pocket (*Figure 2—figure supplement 1B*). All four StART-like domains that expressed as soluble proteins (Ysp2S1 and YspS2 from Ysp2p, Lam4S1 and Lam4S2 from Lam4p) were purified to >95% purity (*Figure 2—figure supplement 1C*), and these proteins all

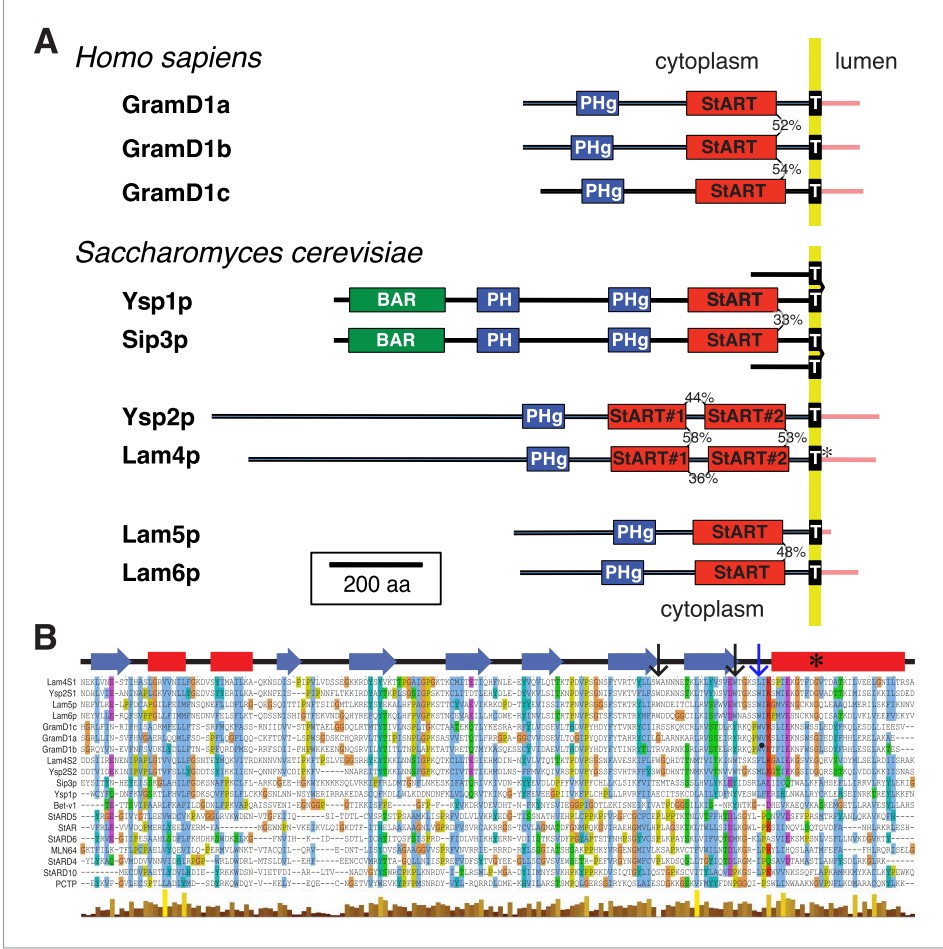

**Figure 1**. A new family of conserved lipid transfer proteins (LTPs) in the StART superfamily. (**A**) StART-like domains are found in the predicted cytoplasmic domains of three human proteins and six yeast proteins, which are three pairs of paralogs. Proportions of identical residues in recently duplicated domains are indicated. Previous identifications were limited to GRAM domains in the pleckstrin homology superfamily (PHg) and transmembrane domains (T, in Lam4p the TMD is weakly predicted*). We identify Bin/amphiphysin/RVS (BAR), and other pleckstrin-homology (PH) superfamily domains. Predicted topology places most regions in the cytoplasm except short luminal regions (pink). Scale bar is 200 aa. (**B**) Alignment of yeast and human StART-like domains with Bet-v1 and seven human StARTs, with CLUSTALX coloring of conserved residues, together with secondary structure (above, sheets—blue arrows, helices—red) and quality of alignment (below). The C-terminal helix contains a glycine residue (*) predicted to interact with the omega-1 loop, hence affecting opening/closing of the lipid binding pocket. Arrows point to tryptophans present in Ysp2S1/2 and Lam4S1/2, black: in all 4 domains, blue: in Ysp2S1 only. A two residue insertion is omitted from the final loop of GramD1b.
The following figure supplements are available for figure 1:

**Figure supplement 1**. A family of StART-like domains in all eukaryotes.
**Figure supplement 2**. Relationships of StART-like domains.

---

produced strong FRET signals with DHE (*Figure 2B* and *Figure 2—figure supplement 1D*). This was not observed with denatured Lam4S2 or with a control protein (soybean trypsin inhibitor) (*Figure 2—figure supplement 1E*, and data not shown). The dissociation constant for binding was estimated at 0.5 μM (±0.1) from a binding curve with DHE added in liposomes, as measured from the FRET signal (*Figure 2C*). All four purified StART-like domains bound both cholesterol (the predominant sterol in mammals) and ergosterol (the predominant sterol in yeast) with similar affinity to DHE, as seen by a reduction in FRET of approximately 50% when DHE was mixed with an

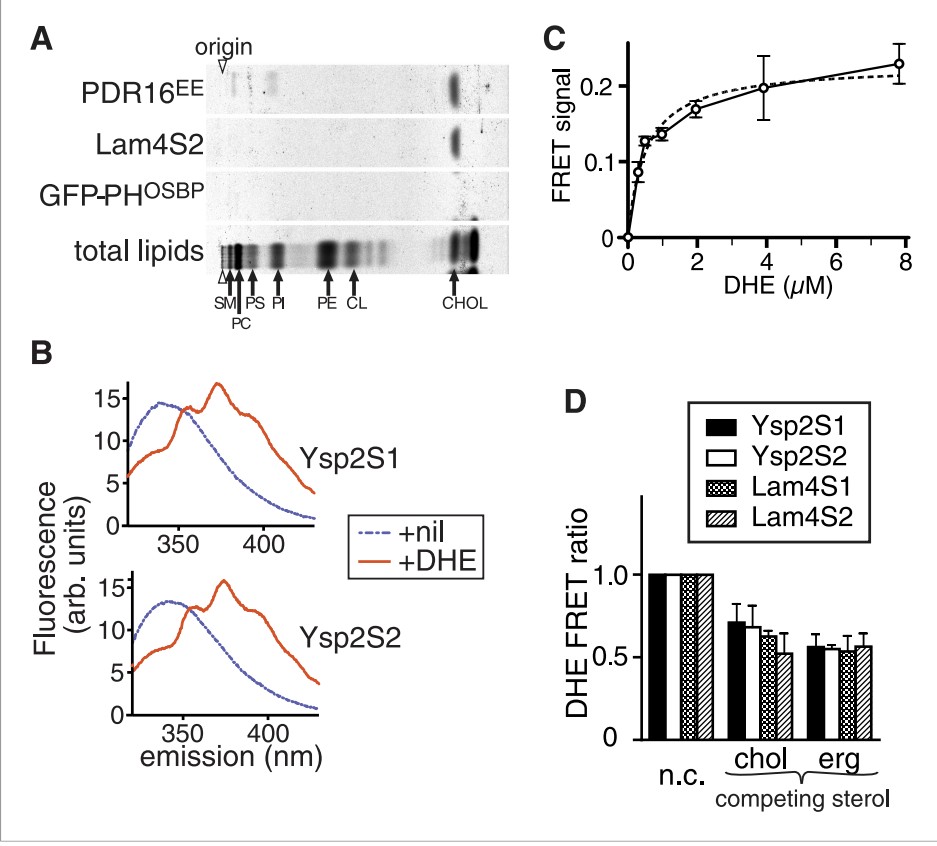

**Figure 2**. StART-like domains in Ysp2p and Lam4p specifically bind sterol. (**A**) The second StART-like domain of Lam4p (Lam4S2) binds cholesterol. Human leukemic cells (HL60) in which all lipids were labelled with 14C-acetate were semi-permeabilized and incubated with bacterially expressed Lam4S2 and two control proteins: Pdr16(EE) which binds cholesterol (*Holic et al., 2014*) and GFP-PH-OSBP (negative control). Lipid extracts of re-isolated proteins were separated by TLC. Positions of major identifiable lipids were ascertained from total lipids (arrows, SM = sphingomyelin, PC = phosphatidylcholine, PI = phosphatidylinositol, PS = phosphatidylserine, PE = phosphatidylethanolamine, CL = cardiolipin). Arrowheads indicate origin. (**B**) FRET between Ysp2S1 or Ysp2S2 and dehydroergosterol (DHE). Tryptophan fluorescence (excitation at 295 nm) with purified protein either on its own or incubated with DHE. (**C**) Tryptophan-DHE FRET of Lam4S2 (1.05 µM) incubated with increasing concentrations of liposomes containing 30% DHE at the final concentrations indicated. The best fitting one saturable site binding curve (dashed line) indicates that Kd for binding = 0.5 µM DHE. (**D**) Effect on DHE FRET signal of adding non-fluorescent sterols (chol—cholesterol, erg—ergosterol) added at the same concentration as DHE and one of four StART-like domains from Ysp2p and Lam4p. 'n.c.' = no competitor, signal defined as 1; lipids added in methanol.

The following figure supplement is available for figure 2:

**Figure supplement 1**. Lipid binding properties of StART-like domains.

equal concentration of non-fluorescent sterol (*Figure 2D*). The sub-micromolar affinity for sterol is similar to other sterol transfer proteins such as Osh4p and Pry1p (*Im et al., 2005*; *Choudhary and Schneiter, 2012*).

From analysis of the chromatography plate in *Figure 2A*, we found that Lam4S2 extracted cholesterol but no other lipids from labelled HL60 cells (relative recovery compared to cholesterol of all identifiable phospholipids ≤0.3%). In contrast, PITPα present in the same experiment extracted phosphatidylcholine (PC) and phosphatidylinositol (PI) but not cholesterol (data not shown). Therefore, the lack of recovery of the major phospholipids, including PC, PI, phosphatidylserine (PS) and phosphatidylethanolamine (PE), by Lam4S2 indicates that if it does bind phospholipids non-specifically (*Schrick et al., 2014*), such binding can only be weak (Kd >100 µM).

## Ysp1p, Ysp2p, Sip3p and Lam4p target punctate ER-PM membrane contacts

To understand the physiological role of the StART-like proteins we investigated their locations. While Ysp2p was detectable when expressed from its own promoter (*Figure 3A*), Sip3p expression from its own promoter was so weak as to be almost undetectable (*Figure 3—figure supplement 1A*), so we used a stronger promoter instead (*Figure 3B*). Both proteins targeted puncta in the cell periphery in wild-type strains; identical patterns were seen in strains lacking the endogenous proteins (data not shown). Their paralogs Lam4p and Ysp1p showed the same pattern, also at low levels of expression (*Figure 3—figure supplement 1B,C*). These peripheral puncta might either be on the PM or on the cortical ER (cER). In contrast to these four peripheral proteins, both Lam5p and Lam6p showed complex intracellular targeting to multiple MCSs, including the NVJ and ER-mitochondrial contacts (*Figure 3—figure supplement 2*). These localizations imply that Ysp1p, Ysp2p, Sip3p and Lam4p are involved in similar or overlapping functions, while Lam5p and Lam6p have a different function.

Because we knew the lipid specificity for Ysp2p and Lam4p, we decided to focus on these proteins as well as Sip3/Ysp1p that are also found in peripheral puncta. First we considered the effect of over-expression. For Ysp2p, high expression had little effect on its distribution (*Figure 3—figure supplement 3A*). However, in cells highly expressing Sip3p, protein accumulated in the ER (*Figure 3B*). This suggests that Sip3p, and by extension Ysp1p, Ysp2p and Lam4p, target a saturable punctate subdomain of the cER. However, additional information is required to make a precise assignment because cER occupies a high proportion of the periphery in yeast, so co-localization with cER could be coincidental. To refine the localization data, we therefore genetically reduced the amount of cER using a strain lacking 90% of cER. In this 'Δtether' strain six proteins involved in ER-PM tethering are deleted, and the proportion of PM with cER is reduced from 40% to 4% with ER accumulation in the center of cells (*Manford et al., 2012*). In Δtether cells GFP-Ysp2p and GFP-Sip3p were still found in peripheral puncta, although these were less numerous compared to wild-type (*Figure 3C* and *Figure 3—figure supplement 3B*). All puncta with GFP-Ysp2p were in close proximity to a strand of ER extending into the periphery, although each ER strand appeared in fewer confocal sections than the associated punctum (*Figure 3C*, arrows). The same was found for Ysp1p, Sip3p and Lam4p (*Figure 3—figure supplement 3C*), indicating that all four proteins target ER-PM contacts that are distinct from those mediated by the six tether proteins previously identified.

We next determined sequence elements required for contact site targeting by analyzing variants of Ysp2p. Surprisingly, we found that Ysp2p is targeted normally to cER puncta in the absence of its N-terminal domains. Unexpectedly, this targeting resulted from just the Ysp2 C-terminal region (Ysp2CT), including the predicted luminal domain (*Figure 3—figure supplement 4*). Similarly, for Ysp1, the C-terminus alone produced punctate targeting (data not shown). Also, because localization to ER-PM contacts is consistent with being embedded either in the PM or in the ER, we induced expression of GFP-Ysp2CT from the *GAL1* promoter after inactivating *SEC18* (yeast NSF). Despite the block in secretion, newly expressed Ysp2CT still localized to peripheral puncta, while a PM resident protein could only attain its normal distribution if expression was induced before inactivating *SEC18* (*Figure 3—figure supplement 5*). This indicates that Ysp2p reaches its final destination without leaving the ER.

Finally we asked if Ysp2p is colocalized with other cER proteins. We first found partial but significant overlap between Ysp2 and Lam4p, its paralog (*Figure 3—figure supplement 6A*). Next, we found a much greater colocalization of Ysp2p with Sip3p with 87% of puncta being double positive (18 cells, 17.2 puncta per cell, s.d. 11%) (*Figure 3D*). This was only true in low expressing cells, as overexpression of Sip3p caused delocalization of Ysp2p from puncta into the ER (*Figure 3E*). For comparison, we examined other punctate cER markers, including the tricalbin Tcb2p, the most punctate of the known ER-PM tethers. Tcb2-GFP at endogenous levels did not significantly colocalize with Ysp2CT (*Figure 3—figure supplement 6B*). Similarly, other punctate cER proteins such as Lnp1p and viral TGBp3 did not colocalize with Ysp2CT (data not shown) (*Wu et al., 2011*; *Chen et al., 2012*). The colocalisation of Ysp2p and Sip3p is therefore specific, and suggests a functionally important relationship between Ysp2p and Sip3p.

In summary, Ysp2p and its homologs are ER proteins strongly enriched at puncta in the cER that may define a sub-class of ER-PM contact site.

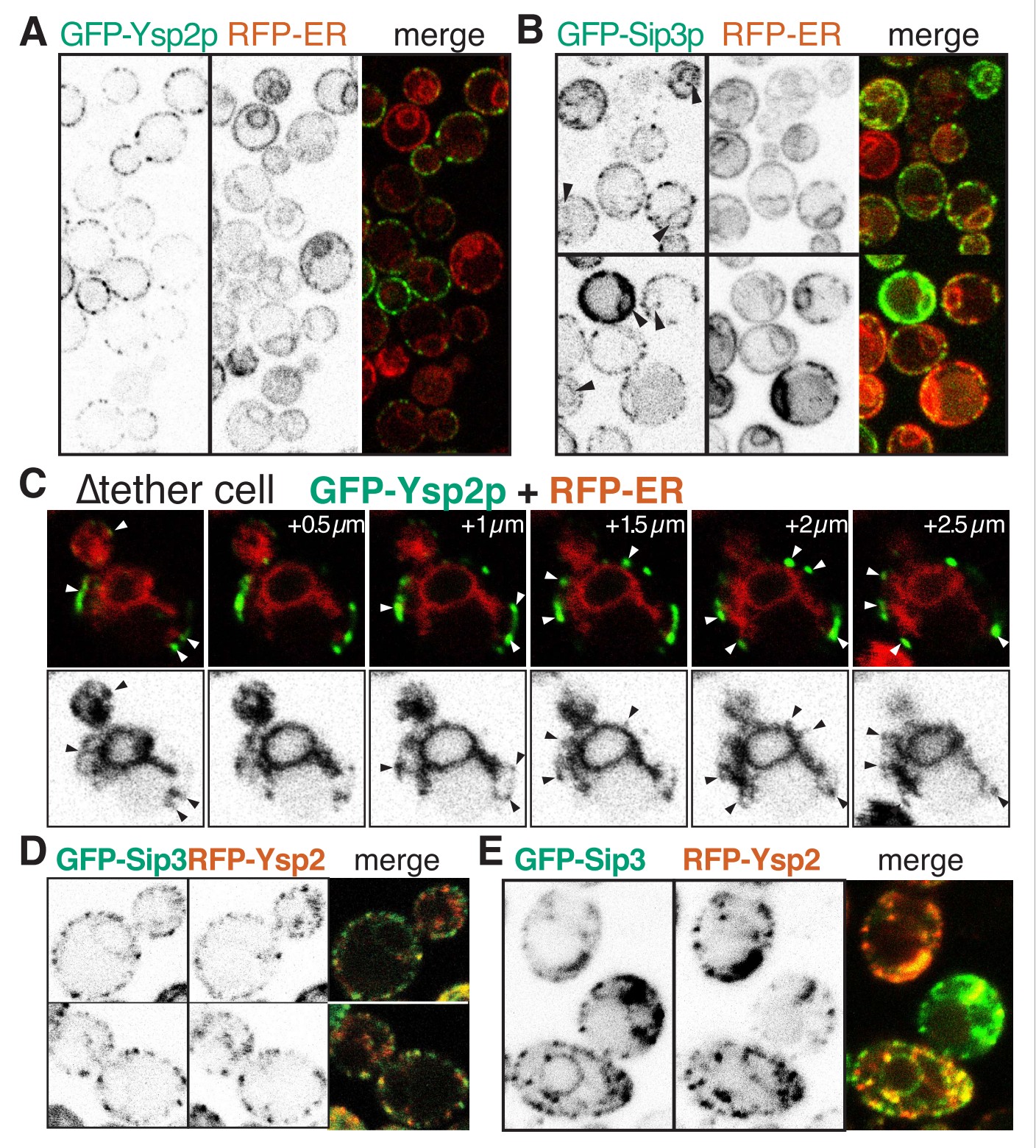

**Figure 3**. GFP-Ysp2p and Sip3 at ER-PM contacts in wildtype and Δtether cells. (**A** and **B**) GFP-Ysp2p and GFP-Sip3p (*PHO5* promoter) in cells co-expressing RFP-ER, showing separate channels (inverted grey-scale) and merges. Arrowheads indicate nuclear envelopes containing GFP-Sip3p. (**C**) Confocal sections of a Δtether cell expressing GFP-Ysp2p and RFP-ER (top), with high contrast inverted grey-scale images of RFP-ER (bottom) to

*Figure 3. continued on next page*

*Figure 3. Continued*

visualize endoplasmic reticulum (ER) strands extending to Ysp2p-positive peripheral puncta (arrowheads). (**D**) Cells co-expressing GFP-Sip3p and RFP-Ysp2 at low levels, with separate channels as inverted grey-scale images and the merge, showing colocalization in most puncta. (**E**) As **D**, but showing cells with high levels of GFP-Sip3p, which accumulates internally with delocalized internal RFP-Ysp2p.

The following figure supplements are available for figure 3:

**Figure supplement 1**. Sip3p Ysp1p and Lam4p expressed from their own promoters.

**Figure supplement 2**. GFP-Lam5p and -Lam6p target the NVJ and ER-mitochondrial contact sites.

**Figure supplement 3**. Ysp2p and its homologs target ER-PM contacts.

**Figure supplement 4**. Analysis of punctate targeting by Ysp2p.

**Figure supplement 5**. Ysp2CT targets peripheral puncta after imposition of *sec18-1* blockade to SNARE-mediated transport.

**Figure supplement 6**. Ysp2 colocalizes partially with Lam4p but not significantly with Tcb2p.

## Ysp2p function correlates with sterol binding to its StART-like domain

Since we know the lipid binding specificity of Ysp2p and Lam4p, we next looked for sterol-related phenotypes associated with loss of these genes. While Δ*lam4* has no known phenotype, Δ*ysp2* induces sensitivity to the polyene antifungals amphotericin B (AmB) and nystatin, which function by extracting ergosterol from membranes (*Anderson et al., 2014*). Δ*ysp2* is 30th most sensitive out of 4130 single gene deletion strains tested with AmB and nystatin, while Δ*ysp1* and Δ*sip3* are even more sensitive (11th and 3rd/4130) (*Hillenmeyer et al., 2008*). Deletion strains that we re-made for the six yeast StART-like proteins accurately reproduced the published sensitivities to polyenes (AmB *Figure 4A* and Nystatin *Figure 4—figure supplement 1A*). Δ*ysp1* and Δ*sip3* were both highly sensitive and Δ*ysp2* was moderately sensitive, while Δ*lam4*, Δ*lam5* and Δ*lam6* were as wildtype. The same pattern of sensitivities was found with natamycin, another polyene (data not shown).

Since loss of Ysp2p causes AmB sensitivity, we used this phenotype to identify which aspects of the protein are critical for its function. Firstly, plasmid-borne GFP-tagged Ysp2p expressed from its own promoter restored AmB resistance (*Figure 4B*), showing that the construct localized to cER puncta is functional. Next, we looked for active domains within Ysp2p. By deleting regions of Ysp2p (*Figure 4—figure supplement 1B*), we found that rescue of AmB-sensitive growth was affected most upon removing either the C-terminal anchor or the second StART-like domain (*Figure 4B*). On the other hand, deletion of Ysp2S1 had no effect. This indicated that both the C-terminus and the second StART-like domain are necessary for Ysp2p function. We next determined which domains are sufficient for rescue. The C-terminus was inactive, even though it is sufficient for punctate cER localization, (*Figure 4C*, *Figure 3—figure supplement 4B*). In contrast, both Ysp2S1 and Ysp2S2 rescued AmB$^s$ growth, although these required over-expression (*Figure 4C*), possibly because the constructs are diffusely cytoplasmic (data not shown). Therefore Ysp2S2 is the only component that is both necessary and sufficient, but it requires overexpression if it is cytoplasmic rather than membrane anchored.

To investigate the activity of YspS2 in rescuing AmB sensitivity, we compared it to other StART-like domains. Activity was found not only in Ysp2S1, but also Lam4S1, Lam4S2, and the StART-like domains of Lam5p and Lam6p (*Figure 4C*). Significantly, human GramD1a increased resistance to AmB in both mutant and wild-type cells. The conservation of activity from yeast to human strongly indicates that the StART-like domains are the active portions of this protein family. Furthermore, constructs that rescued Δ*ysp2* generally also rescued Δ*ysp1* and Δ*sip3* (*Figure 4—figure supplement 1C*), which indicates that there may be considerable overlap in function between these homologs.

To investigate the role of sterol binding, we made a non-sterol-binding mutant by mutating a conserved glycine residue in the C-terminal helix predicted to interact with the omega-1 loop (*Figure 1B* and *Figure 2—figure supplement 1B*). Replacing G1205 in Ysp2p with alanine, threonine

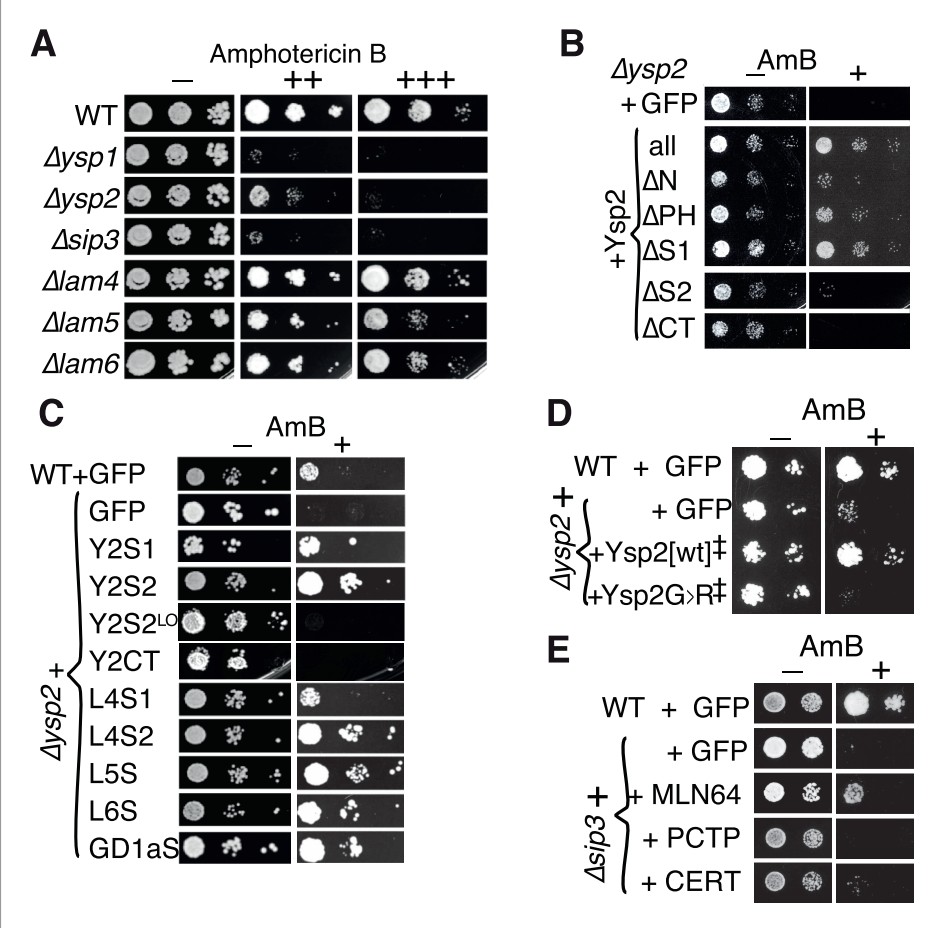

**Figure 4**. Amphotericin B (AmB[S]) phenotypes and rescue by StART-like domains that bind sterol. (**A**) Dilutions of cells with single gene deletions were compared with the wild-type parental strain (WT) for ability to grow at two concentrations (moderate and high) of AmB, with AmB = 0 to control for cell number. (**B**) Effect of AmB[S] growth by Δ*ysp2* cells from GFP-Ysp2p under its own promoter with individual domains deleted. (**C**) Rescue of Δ*ysp2* by domains of Ysp2p (S1/S2/CT), Lam4S1/S2, and StART-like domains of Lam5p Lam6p and human GramD1a. All plasmids had the *PHO5* promoter for moderately high expression, except Y2S2[LO] had the weakly expressing *YSP2* promoter. (**D**) Growth by Δysp2 cells expressing Ysp2p or the G > R mutant under the *YSP2* promoter ± AmB. ‡ indicates that Ysp2S2 in both plasmids was substituted by Lam4S2 (53% identical, 72% homologous). (**E**) Growth ± AmB by Δ*sip3* cells overexpressing human LTP domains of MLN64, PCTP and ceramide transfer protein (CERT) (*TPI1* promoter).

The following figure supplement is available for figure 4:

**Figure supplement 1**. Activities of different StART-like domains.

---

and arginine successively reduced the ability of Ysp2S2 to rescue AmB-sensitive growth (data not shown). Mutating the equivalent residue in Lam4S2 (G1119R) completely inhibited sterol binding in vitro (**Figure 4—figure supplement 1D**), showing that this residue is indeed critical. Substitution of this G → R mutated domain abolished rescue of AmB sensitivity by Ysp2p (**Figure 4D**), supporting a correlation between Ysp2p function and sterol binding by Ysp2S2. We next tested if the AmB-sensitive phenotype of Δ*ysp2* cells could be rescued by a heterologous StART domain with very little sequence similarity to Ysp2S2, as was previously done for PITP/Sec14 (**Skinner et al., 1993**). For this we expressed StART domains of human MLN64 (sterol-specific), PCTP (phosphatidylcholine-specific) and CERT (ceramide-specific) (**Figure 1B** and **Figure 1—figure supplement 2B**). The StART domain of MLN64 produced significant rescue, but the StART domains of PCTP and CERT did not (**Figure 4E**). This strongly supports the idea the function of Ysp2p resides in its interaction with sterol, rather than a specific protein–protein interaction.

## Ysp2p function correlates with the ability of its StART-like domain to reach across ER-PM contacts

Without its C-terminus, Ysp2p loses both targeting and function (*Figure 3—figure supplement 2* and *Figure 4C*). To examine the importance of anchoring further, we used the AmB resistance assay to determine the role of intracellular localization of Ysp2S2 by anchoring it to the cytoplasmic face of different organelles (*Figure 5A*). Ysp2S2 activity was completely inhibited when anchored to either mitochondria or the vacuole. It was slightly activated by anchoring throughout the ER, and greatly activated by anchoring either to ER-PM contacts (with Ysp2CT) or to the PM as a whole. Thus, Ysp2S2 is activated by anchoring to either side of ER-PM contacts, especially to the contacts themselves, but inhibited by anchoring elsewhere.

Ysp2p is anchored to the ER membrane, which at contact sites is separated from the PM by a gap of ~30 nm (range 15–60 nm) (*West et al., 2011*). This distance is large compared to the size of the StART-like domain itself (diameter 2.5 nm). While Ysp2p may act to extract or sense ER sterol, that is, function *in cis*, we wondered if its function requires it to act *in trans* by reaching across ER-PM contacts. Since unstructured peptide chains can extend up to 0.38 nm per residue (*Pillardy et al., 2001*), a linker between Ysp2S2 and a TMD anchor would need to be ≥40 aa to allow Ysp2S2 to cross the minimal 15 nm gap. We expressed constructs with Ysp2S2 linked to the generic ER anchor by unstructured linkers of varying length. Linkers of up to 10 aa (≤4 nm) completely inhibited Ysp2S2. In contrast, YspS2 was active when the linker was 40 aa (≤15 nm) or longer (*Figure 5B*). More stringent testing, achieved using a higher concentration of AmB, showed that the 40 aa linker was only partially active, and that Ysp2S2 with linkers of 71 aa (≤27 nm) and 103 aa (≤39 nm) was progressively more active (*Figure 5C*). This suggests that Ysp2p is only active when its StART-like domain can reach across the ER-PM contact to the other side.

## Elimination of Ysp1p, Ysp2p and Sip3p slows transport of exogenously supplied sterols to the ER

We have shown that (i) Ysp2p is localized to puncta in the cER, (ii) Ysp2p can bind ergosterol via its StART-like domain, (iii) the function of Ysp2 and related genes can be partially rescued by an evolutionarily distant sterol transfer domain, and (iv) expression of Ysp2p variants can rescue the AmB-sensitivity of Δysp2 cells only when the StART-like domain is attached to a linker capable of spanning the ER-PM contact distance. Thus Ysp2p is positioned either to mediate sterol exchange between the ER and PM at a MCS or to act as a sterol-sensor that transmits information on lipid composition/organization between the ER and PM. Although we know less about the other members of the yeast StART-like family, we have established that Sip3p and Ysp1p localize to peripheral ER puncta and have overlapping functions, so we are interested in these other proteins alongside Ysp2p.

First we measured the anterograde transport of ergosterol between the ER and PM (*Georgiev et al., 2011*). This is a direct transport assay that measures the appearance in the PM of a pulse of radiolabelled ergosterol generated in the ER. We metabolically labeled cells with [$^3$H-methyl]-methionine for 4 min and either stopped the reaction immediately or chased the cells in excess unlabelled methionine for 15 min. At both time points we measured the ratio of radioactive ergosterol to total ergosterol (specific radioactivity, or SR) in purified PM fractions (SR$_{PM}$) and in unfractionated extracts (SR$_{cell}$). The amount of newly synthesized sterol in the PM was expressed as the relative specific ratio (RSR) = SR$_{PM}$: SR$_{cell}$. At long time points (90 min) RSR typically tends to 1, but even by the end of the pulse, RSR was 0.38 (s.d. 0.035, n = 5) in wild-type cells, which is similar to values achieved previously (*Georgiev et al., 2011*). RSR increased to 0.61 (s.d. 0.072, n = 5) after 15 min chase. Deletion of *YSP2* alone produced no significant change in traffic during the pulse or after the chase (0.33 and 0.63, range for both ±0.032, n = 2). This results rules out that Ysp2p is the sole transporter of sterol between ER and PM. However, because the assay has a rapid first phase (pulse period >t1/2), we cannot determine whether loss of Ysp2p causes a partial defect that would be undetectable in our protocol.

To test for more subtle effects that Ysp2p and homologs might play in sterol transport, we deployed two assays to measure the transport of exogenously supplied sterols (DHE and cholesterol) from the PM to the ER (*Georgiev et al., 2011*). Both assays require cells to be permissive for sterol uptake, which was achieved for DHE by reversibly inducing hypoxia (*Georgiev et al., 2011*), and for cholesterol by using *upc2-1* strains (*Crowley et al., 1998*). Key mechanistic steps that generate the eventual read-out of the assays are the insertion of sterol into the PM (requiring the ABC transporters

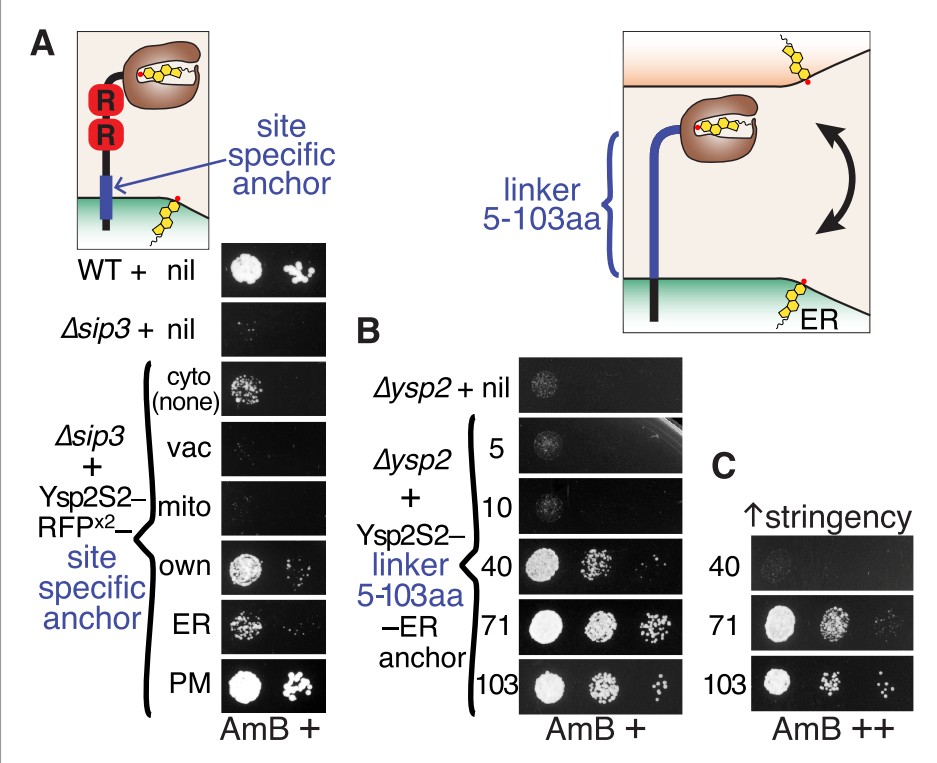

**Figure 5**. Ysp2S2 is activated by anchoring that allows crossing of ER-PM contacts. (**A**) Anchoring at different locations. AmB[s] growth of Δ*sip3* cells transformed with plasmids weakly expressing (*YSP2* promoter) Ysp2S2-RFP (dimeric) followed either by nothing (cytoplasmic) or by anchors for specific sites: vacuole (Nyv1p, all), mitochondria (Tom6p, all), own anchor (Ysp2 1249–1438), generic ER (Scs2: 220–244), and plasma membrane (PM) (Sso1p, all). WT and Δ*sip3* cells carrying empty plasmids were included as controls. (**B** and **C**) Varying the length of the linker. AmB[s] growth of Δ*ysp2* cells carrying an empty plasmid or weakly expressing (*YSP2* promoter) Ysp2S2 anchored to the ER by the TMD of Scs2 (residues 226–244) with intervening linkers of 5, 10, 40, 71 or 103 residues. Cells in **B** were grown on plates with moderate AmB (63 ng/ml AmB). In **C**, cells with 40, 71 and 103 aa linker constructs were grown at higher stringency (125 ng/ml AmB). Diagrams indicate the variable sites for each experiment in blue. Equal cells were plated, as shown on control plates (*Figure 5—figure supplement 1*).

The following figure supplement is available for figure 5:

**Figure supplement 1**. Controls for tests of LAM domain function at ER-PM contacts.

Aus1p and Pdr11p), non-vesicular movement from PM to ER, and esterification at the ER requiring the ACAT enzymes Are1p and Are2p (*Figure 6A*) (*Raychaudhuri et al., 2006*).

We first assayed the rate of DHE esterification in wild-type cells and single delete strains. DHE was loaded into the PM under hypoxic conditions and transport was initiated by exposing the cells to air. DHE esterification commenced after an initial lag period of 1 hr that was common to all strains tested. The rate of DHE esterification was reduced by ∼40–50% in all of Δ*ysp1*, Δ*ysp2* and Δ*sip3* cells, but not affected by Δ*lam4*, Δ*lam5* or Δ*lam6* (*Figure 6B* and data not shown). We next assayed cholesterol import in the *upc2-1* strain background. The use of this background avoided the lengthy hypoxic induction used for DHE uptake, and because cholesterol is poorly tolerated in the yeast PM its flux through the import pathway leading to esterification is more rapid than for DHE (*Li and Prinz, 2004*). Compared with the *upc2-1* parental strain, the rate of cholesterol esterification was reduced by 30–35% in both Δ*ysp2* and Δ*sip3* single deletes (*Figure 6C*). This indicates that similar defects exist for retrograde traffic of both exogenous sterols we tested.

To rule out trivial explanations for these results, we verified that Aus1p and Pdr11p localized similarly in deletion and wild-type cells (data not shown) (*Li and Prinz, 2004*). We also found that sterol esterification activity was not altered by deletion of Ysp2p and was marginally increased by loss of Ysp1p or Sip3p

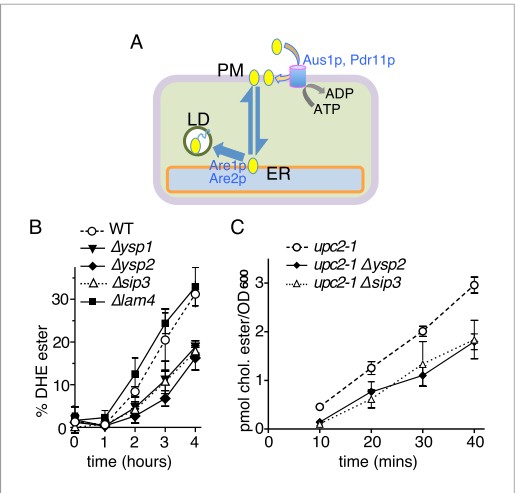

**Figure 6**. Retrograde sterol traffic is slower in strains lacking Ysp1p Ysp2p or Sip3p. (**A**) Diagram of retrograde traffic pathway for exogenous sterols. Apart from the hypothesized sterol transfer protein (double headed arrow), other steps include insertion into the PM by ABC transporters Aus1p and Pdr11p, and esterification in the ER by ACAT enzymes Are1p and Are2p prior to storage in lipid droplets (LD). (**B**) Retrograde traffic of DHE in four single delete strains of the yeast StART-like family were compared to wildtype controls. DHE ester formation was followed by HPLC during redistribution of DHE away from the PM. (**C**) Retrograde traffic of cholesterol in wildtype and two delete strains ($\Delta ysp2$ and $\Delta sip3$) as determined from esterification of cholesterol added exogenously. Strains were created in a WPY361 (*upc-1*) background to allow cholesterol uptake.

The following figure supplement is available for figure 6:

**Figure supplement 1**. Strains lacking Ysp1p Ysp2p or Sip3p have no significant changes in sterol handling.

(*Figure 6—figure supplement 1A*). Another issue we investigated was whether loss of StART-like proteins altered overall ergosterol distribution, which could interfere with the kinetics of our assays (*Li and Prinz, 2004*). Firstly, we measured the total amount of ergosterol in gradient fractions enriched for PM; this was not significantly changed in the delete strains (*Figure 6—figure supplement 1B*). Secondly, we assessed the organization of PM sterol, since changes in other lipids, especially sphingolipids, might lead to increased traffic of endogenous sterol that would compete with exogenous lipid (*Simons and Ikonen, 2000*; *Li and Prinz, 2004*; *Das et al., 2014*). The proportion of sterol that partitions into detergent–insoluble complexes was unaffected by $\Delta ysp1$, $\Delta ysp2$ or $\Delta sip3$ deletions (54 ± 1%, *Figure 6—figure supplement 1C*). Therefore, effects on the distribution and organization of ergosterol do not explain the effect of $\Delta ysp1$, $\Delta ysp2$ and $\Delta sip3$ on esterification of exogenous sterol. This leaves one major site of action of these proteins: they appear to act on the retrograde transport step itself, although we have not tested if their mechanism of action is direct or indirect.

## Discussion

We discovered a StART-like family of membrane-anchored lipid-binding proteins that is conserved throughout eukaryotes. Among the six yeast family members, the StART-like domains of Ysp2p and Lam4p specifically bind sterols, and the StART-like domains of GramD1a, Lam5p and Lam6p rescue AmB[s] growth, indicating that these too bind sterol. We were not able to test the StART-like domains of Ysp1p and Sip3p as they were poorly expressed, but since the AmB[s] phenotype of $\Delta ysp1$ and $\Delta sip3$ cells could be rescued by sterol-specific StART-like domains (*Figure 4—figure supplement 1B*), it appears that these proteins also bind sterols. Thus, all members of this new protein family in yeast and humans may be specific for sterol.

The presence of TMDs is a key aspect for the entire StART-like family. Ysp1p, Ysp2p, Sip3p and Lam4p reside in puncta in the cER network that may define a novel class of ER-PM contact sites, while Lam5p and Lam6p target internal MCSs. This suggests that the fundamental molecular feature for members of this family is being a **l**ipid transfer protein **a**nchored at a **m**embrane contact site (hence 'LAM'). Previous studies of Ysp1p, Ysp2p and Sip3p antedated the prediction of StART-like domains (*Lesage et al., 1994*; *Pozniakovsky et al., 2005*; *Sokolov et al., 2006*), and we suggest providing a uniform nomenclature in the future by renaming these Lam1-3p respectively. Little is known about the human proteins, except that variation at the GramD1b locus is linked to lymphoma/leukemia (*Di Bernardo et al., 2008*; *Conde et al., 2010*), and we suggest renaming GramD1a-c as hLAMa-c. Intriguingly, the one LAM gene to have previously been linked to lipids is *LAM4*, which we previously identified in a random transposon screen as one of 20 genes involved in sterol import (*Sullivan et al., 2009*). Although $\Delta lam4$ has no sterol traffic defect, this might be explained by its lower level of expression compared to Ysp2p, so the previous results suggest that transposon insertion in *LAM4* can produce a dominant negative effect.

Ysp1p and Sip3p are outliers in terms of sequence to the other yeast and human StART-like proteins (*Figure 1—figure supplements 1, 2*). Nevertheless, the co-localization of Sip3p with Ysp2p, and the fact that altered levels of Sip3p cause redistribution of Ysp2p suggest that these proteins interact. It is possible that they target a specific class of ER-PM contact site, similar to the sub-division of ER-mitochondria contacts into regions with different molecular components (*Lahiri et al., 2014*). However targeting to puncta by Ysp2p does not appear to require stoichiometric interaction with Sip3p, as highly overexpressed Ysp2p remains punctate (*Figure 3—figure supplement 3A*). Furthermore, Δ*sip3* has a much greater polyene sensitivity than Δ*ysp2*, even though the two mutants have similar retrograde sterol defects, which indicates that the these proteins have significant non-overlapping functions.

We focused on studying the physiological role of Ysp2p. The activity of the StART-like domain of Ysp2S2 was maximized by anchoring it to either PM or ER with a linker of ≥100 aa. With a linker of 10 aa (≤4 nm) it was completely inactive, even though this is long enough for the domain to access lipids *in cis* (*Schulz et al., 2009*). Therefore, when anchored in the ER, Ysp2S2 function appears to function *in trans*. The activity of Ysp2S2 increased as the linker was extended from 40 aa (≤15 nm) to 103 aa (≤39 nm). However, linkers in the StART-like proteins are 55–75 aa (*Figure 1A*), similar to TULIPs (45–75 aa), which are also membrane anchored (*Toulmay and Prinz, 2012*; *Schauder et al., 2014*). Since the natural linkers are shorter than those we needed for maximal activation of artificial Ysp2S2–TMD constructs, other factors, such as the accessory domains found commonly in this family, may contribute to activation in vivo. Hydrophobic loops of C2 domains and of PH-like domains penetrate into membranes to destabilize them (*Ramachandran et al., 2009*; *Paddock et al., 2011*), which may disturb the local bilayer and contribute to function.

An important point to make is that we have not yet established how Ysp1p, Ysp2p and Sip3p modulate the movement of sterol between the PM and ER. Their mode of action might be indirect and regulatory, or they themselves might directly mediate sterol transfer. This lack of certainty arises because our assays of retrograde transport can be affected by sterol partitioning within the PM (*Li and Prinz, 2004*). A key experimental question for the future will be to develop a new biophysical approach that shows how many sterol molecules that traffic between PM and ER actually pass into (and out of) the pocket of one of these LTPs. We have also not established how polyene sensitivity arises in cells lacking the StART-like proteins. Studies in pathogenic fungi (*Vincent et al., 2013*) and in sphingolipid mutants (*Sharma et al., 2014*) suggest that AmB sensitivity might be accompanied by gross biophysical change in sterol levels or extractability respectively, but this was not found. Mutations of several different *ERG*osterol biosynthetic genes in pathogenic fungi induce clinically relevant polyene resistance that is mediated by a range of stress responses to reduced sterol (*Vincent et al., 2013*). However the situation may be different in *S. cerevisiae* where deletions of single non-essential *ERG* genes do not induce polyene resistance, and one deletant is sensitive to both AmB and nystatin (Δ*erg4*: 38th out of 4130 deletion strains tested) (*Hillenmeyer et al., 2008*). As for altered sphingolipid metabolism, which can affect sterol traffic and AmB sensitivity (*Li and Prinz, 2004*; *Das et al., 2014*; *Sharma et al., 2014*), this has been linked to Pmp3p, a conserved small hydrophobic PM protein of unknown function (*Huang et al., 2013*; *Bari et al., 2015*). However, Pmp3 only affects sensitivity to AmB not other polyenes, so it seems likely that LTPs anchored at MCSs, mutants of which are sensitive to all polyenes, act in a separate pathway.

## Materials and methods

### Chemicals

Unless otherwise stated chemicals were obtained from Sigma–Aldrich, UK. Lipids were obtained from: Avanti Polar Lipids, Alabaster, Alabama (cholesterol, DOPE); Lipid Products, UK (PC, PS); MP Biomedicals, Santa Ana, California (ergosterol) and Sigma (DHE).

### Plasmids and strains

All plasmids are listed in *Table 1*. Deletion strains were obtained either from freezer stocks of the yeast deletion collection (BY4741/2, and *KanMX*) or were made by the PCR method with heterologous markers, as listed in *Table 2*.

**Table 1**. Plasmids used in this study

| Bacterial expression | Constructs all start: MGGSHHHHHHGMASHHHHHARA |
|---|---|
| His-Ysp2S1 | + PVMT + Ysp2 829-1027 + R |
| His-Ysp2S2 | + PVMT + Ysp2 1027-1244 + R |
| His-Lam4S1 | + M + Lam4 731-938 + DV |
| His-Lam4S2 | + M + Lam4 946-1155 + DV |
| His-Lam4S2(G > R) | as His-Lam4S2, G1119 > R |
| Yeast expression | unless stated pRS series, 405 = integrating *LEU2*, 406 = integrating *URA3*, 316 and 416 = CEN *URA3*; **unless stated PHO5 promoter {168}** |
| GFP-Ysp2p | 416: GFP + Ydr326c/Ysp2p ORF [1438], with 2 extra residues to facilitate cloning in five sections: K828 > KT and S1244 > SR |
| GFP-Ysp1p/Sip3p/Lam4p | 416: GFP + whole ORFs: Yhr155w/Ysp1p [1228], Ynl257c/Sip3p [1229], Yhr080cp/Lam4p [1348] |
| GFP-Lam5/6p* | 406: GFP + whole ORFs: Yfl042cp/Lam5p M1 > S [674], Ylr072wp/Lam6p M1 > S [693] |
| GFP-Ysp1p/Ysp2p/Sip3p/Lam4p: own promoters | as above for GFP-ORF, except replace *PHO5* promoter with: *YSP1* {951}, *YSP2* {459}, *SIP3* {360}, *LAM4* {701} |
| RFP-ER | 405: dimeric dsRed tdimer2(12) [464] + RNSKP (linker) + ENESSS•MGIFILVALLILVLGWFY•R = Scs2p 220–244 = linker[6] + TMD[18] + lumen[1] |
| RFP-Tom6p | as RFP-ER, but after RNSKP: G + Tom6p [61] |
| RFP-Ysp2p | 406: dimeric dsRed tdimer2(12) [464] + Ysp2 K828 > KT and S1244 > SR |
| GAL > GFP-Ysp2p | as for GFP-Ysp2p, except 406, and replace *PHO5* promoter with *GAL1* {807} |
| GFP-Ysp2ΔN | 416 *YSP2* prom.: GFP + Ysp2 611-1438 |
| GFP-Ysp2ΔPH | 416 *YSP2* prom.: GFP + Ysp2 1-610 + 829-1438 |
| GFP-Ysp2ΔS1 | 416 *YSP2* prom.: GFP + Ysp2 1-828 + T + 1028-1438 |
| GFP-Ysp2ΔS2 | 416 *YSP2* prom.: GFP + Ysp2 1-1027 + R + 1245-1438 |
| GFP-Ysp2S1S2CT | 416 *YSP2* prom.: GFP + Ysp2 829-1438 |
| GFP-Ysp2S2CT | 416 *YSP2* prom.: GFP + Ysp2 1028-1438 |
| GFP-Ysp2CT | 416 *YSP2* prom.: GFP + Ysp2 1245-1438 |
| GFP-Ysp2ΔextremeC | 416 *YSP2* prom.: GFP + Ysp2 1-1319 |
| GFP-Ysp2ΔCT | 416 *YSP2* prom.: GFP + Ysp2 1-1246 |
| RFP-Ysp2CT | as RFP-ER, but after RNSK: LGSAPVMSR + Ysp2 1245-1438 |
| GFP only | 416: GFP + GFP |
| GFP-Ysp2S1 | 416: GFP + Ysp2 829-1028 |
| GFP-Ysp2S2 | 416: GFP + Ysp2 1027-1244 + R |
| GFP-Lam4S1 | 416: GFP + Ysp2 759-929 |
| GFP-Lam4S2 | 416: GFP + Lam4 968-1140 |
| GFP-Lam5S | 416: GFP + Lam5 381-586 + AS |
| GFP-Lam6S | 416: GFP + myc tag + Lam6 374-582 + DV |
| GFP-GramD1aS | 416: GFP + GramD1a 359-547 + DV |
| GFP-Ysp2S2sw4 | 416 *YSP2* prom.: GFP + Ysp2 1-1044 + Lam4 S953-V1161 + SR + Ysp2 1245-1438 |
| GFP-Ysp2S2sw4G > R | as GFP-Ysp2S2sw4 G1119 > R |
| GFP-StART-MLN64 | 406: *TPI1* prom. {412}: GFP + MLN64 216-445 |
| GFP-StART-PCTP | 406: *TPI1* prom.: GFP + PCTP [214] |
| GFP-StART-CERT | 406: *TPI1* prom.: GFP + CERT$_L$ 397-624 |

*Table 1. Continued on next page*

*Table 1. Continued*

| Ysp2S2-RFP | 405 *YSP2* prom.: M + Ysp2 1022-1244 + 40 aa linker + RFP + 27 aa linker + RFP + HLFLRNSK + final 8aa: LGSQSMFD |
|---|---|
| Ysp2S2-RFP-Vacuole | as Ysp2S2-RFP; final 8 aa → PGASYQ + Nyv1p [253] |
| Ysp2S2-RFP-Mito | as Ysp2S2-RFP; final 8 aa → PG + Tom6p [61] |
| Ysp2S2-RFP-own CT | as Ysp2S2-RFP; final 8 aa → LGSAPVMSR + Ysp2 1245-1438 |
| Ysp2S2-RFP-ER | as Ysp2S2-RFP; final 8 aa → P + Scs2p 220-244 (see RFP-ER, above) |
| Ysp2S2-RFP-PM | as Ysp2S2-RFP; final 8 aa → PGAS + Sso1p [290] |
| Ysp2S2-5aa-ER | 405 *YSP2* prom.: M + Ysp2 1057-1224 + linker SRLE + Scs2 225-244 = linker [1] + TMD[18] + lumen [1] (S•MGIFILVALLILVLGWFY•R). Total linker = 5 aa |
| Ysp2S2-10aa-ER | as Ysp2S2-5aa-ER; after SRLE insert extra 5 aa: Scs2 220-224 |
| Ysp2S2-40aa-ER | as Ysp2S2-5aa-ER; after SRLE insert extra 35 aa: Scs2 190-224 |
| Ysp2S2-71aa-ER | as Ysp2S2-5aa-ER; after SRLE insert extra 66 aa: Scs2 159-224 |
| Ysp2S2-103aa-ER | as Ysp2S2-5aa-ER; after SRLE insert extra 98 aa: KL + Scs2 129-224 |
| Pdr11p-GFP† | pWP1251. 316: *PDR11* prom.: Pdr11p-GFP |
| Aus1p-GFP† | pWP1220. 316: *AUS1* prom.: Aus1p-GFP |

Cloned regions are from *S. cerevisiae* (S288c) or human I.M.A.G.E. clones; size of whole proteins/promoters are in brackets, for protein [aa] and DNA {bp}; promoters are regions of genome just prior to open reading frame starts; ranges within proteins are not in brackets; changes in natural residues are written as 'X123 > Z'; specific amino acid sequences (e.g., linkers) are underlined.

Plasmids were received as kind gifts from

*Sean Munro.

†Will Prinz.

## Bioinformatics

HHpred (at toolkit.tuebingen.mpg.de) was used online and seeded with sequences of StART domains only (170–210 aa) using HHblits and with maximum accuracy alignment on. HHblits was used with the following settings: 8 rounds, E-value threshold for inclusion = 0.01 (*Remmert et al., 2012*). For the alignment with StARTs, sequences aligned by HHpred seeded with Ysp4S2 were exported into JalView, and edited by hand to eliminate redundant and incomplete sequences and add the sequence of GramD1b, which is missing from the database. For tree drawing, 888 StART-like sequences were reduced to 143 using a non-redundancy filter (PISCES at dunbrack.fccc.edu/) and then the best tree inferred with PHYML (http://www.trex.uqam.ca/) (*Guindon and Gascuel, 2003*). For structural modelling, the sequence of Lam4S2 was submitted to SAM-T08 (*Karplus, 2009*).

Joint sensitivity of strains to AmB and nystatin was obtained by rank ordering the 4130 deletants that were tested with both drugs by *Hillenmeyer et al. (2008)* according to the sensitivities to the two different drugs (data downloaded from Yeast Fitness Database at http://chemogenomics.pharmacy.ubc.ca/fitdb/fitdb2.cgi).

## Lipid binding in HL60 cells

Binding to cellular lipids was carried out as described (*Holic et al., 2014*). In summary, $10^7$ HL60 cells were labelled with 1 µCi/ml [$^{14}$C]-acetate and then permeabilized with Streptolysin-O, prior to incubation with 200 µg of the indicated His-tagged proteins at 37°C for 20 min, recapture of proteins on nickel beads, and then extraction of total lipids. Lipids were resolved by thin layer chromatography on a Whatman Silica Gel 60 TLC plate developed using chloroform:methanol:acetic acid:water 75:45:3:1 (vol/vol). The chromatogram was imaged using a Fuji PhosphorImager screen. Bands that migrated near the solvent front were scraped and re-chromatographed using hexane:diethylether:acetic acid 155:45:2 (vol/vol). For data analysis in ImageJ, background was estimated from the adjacent negative control lane loaded with repurified GFP-PH$^{OSBP}$.

**Table 2.** Strains used in this study

| Strain | Genotype | Reference |
|---|---|---|
| BY4741 | (Euroscarf) MATa his3Δ1 leu2Δ0 met15Δ0 ura3Δ0 | Euroscarf |
| Δysp1 | BY4741 YHR155W::KANMX4 | Euroscarf |
| Δysp2 | BY4741 YDR326C::KANMX4 | Euroscarf |
| Δsip3 | BY4741 YNL257C::KANMX4 | Euroscarf |
| Δlam4 | BY4741 YHR080C::KANMX4 | Euroscarf |
| Δlam5 | BY4741 YFL042C::KANMX4 | Euroscarf |
| Δlam6 | BY4741 YLR072W::KANMX4 | Euroscarf |
| WPY361 (upc2-1) | MATa upc2-1 ura3-1 his3-11,-15 leu2-3,-112 trp1-1 | (Li and Prinz, 2004) |
| Upc2-1 Δysp2 | WPY361 (upc2-1) YDR326C::KanMX4 | This study |
| Upc2-1 Δsip3 | WPY361 (upc2-1) YNL257C::HYGRO$^R$ | This study |
| RS453C | MATα ade2-1 his3-11,15 ura3-52 leu2-3112 trp1-1 | (Levine and Munro, 2001) |
| Δysp1 | RS453C YHR155W::HIS5 S.p. | This study |
| Δysp2 | RS453C YDR326C::HIS5 S.p. | This study |
| Δsip3 | RS453C YNL257C::HIS5 S.p. | This study |
| Δlam4 | RS453C YHR080C::HIS5 S.p. | This study |
| Δlam5 | RS453C YFL042C::HIS5 S.p. | This study |
| Δlam6 | RS453C YLR072W::HIS5 S.p. | This study |
| Tcb2-GFP | RS453C TCB2-EGFP::HIS5 S.p. | This study |
| SEY6210 | MATa leu2-3,-112 ura3-52 his3Δ200 trp1Δ901 lys2-801 suc2-Δ9 GAL | (Manford et al., 2012) |
| ANDY198 (Δtether) * | SEY6210 ist2Δ::HISMX6 scs2Δ::TRP1 scs22Δ::HISMX6 tcb1Δ::KANMX4 tcb2Δ::KANMX46 tcb3Δ::HISMX6 | |
| RSY271 sec18-1† | MATa sec18-1 ura3-52 his4-619 | (Novick et al., 1980) |
| Lnp1-GFP‡ | SFNY 2092, MAT a, ura3-52, leu2-3,112, his3Δ200, LNP1-3xGFP::URA3 | (Chen et al., 2012) |

Kind gifts of strains:
*Chris Stefan.
†Mike Lewis.
‡Susan Ferro-Novick.

## Protein expression and purification

Polyhistidine tagged proteins were expressed on plasmids based on pTrcHis (Life Technologies) in BL21(DE3) *Escherichia coli*, which were induced at 37°C with 0.2 mM IPTG for 6 hr. After lysis in 25 mM TrisHCl pH 8.0, 300 mM NaCl, 5 mM imidazole, 1× complete EDTA-free protease inhibitor cocktail (Roche) and clearing by centrifugation, lysate was loaded onto Ni-NTA agarose beads and washed repeatedly before eluting with the same buffer with 300 mM imidazole. Eluted protein was desalted into 20 mM PIPES, 137 mM NaCl, 3 mM KCl, pH 6.8 and evaluated by Coomassie staining of SDS-PAGE gels. All preparations were >95% pure.

## Tryptophan/DHE FRET

For binding to DHE alone, lipid was pre-incubated with fivefold molar excess of methyl β-cyclodextrin prior to mixing with protein. For competitive binding between DHE and other sterols, lipids (3 μM) were mixed together in methanol and added to protein (2 μM), with a final methanol concentration of 0.2%. Titration of DHE binding to Lam4S2 (WT and G1119R, 1.1 μM) was carried out by incubation

with 100 nm diameter liposomes composed of 40% PC, 20% DOPE, 10% PS and 30% DHE. In all cases protein was incubated with lipids for up to 30 min and fluorescence measured with excitation of 295 nm (slit 5 nm). The FRET response was calculated after deducting DHE alone background, as the ratio between the DHE emission peak (373 nm) and the tryptophan emission peak (340 nm). Curve fitting was carried out using Prism software.

## Microscopy

Cells were grown at 30°C to mid log phase in synthetic medium +2% dextrose containing the appropriate amino acids and bases for plasmid maintenance. Live cells were immobilized between slide and coverslip, and visualized on a confocal microscope (either Leica AOBS SP2 or Zeiss LSM 700) at room temperature. Two color images were obtained using line-by-line switching, and single labelled cells with full range of GFP/RFP expression were used to ensure lack of bleed through between channels.

## Growth assay

Sensitivity to AmB was determined by spotting yeast from log-phase cultures at 20-fold dilutions (e.g., 2000/100/5 cells per spot) on freshly made up plates with AmB added from a 250 µg/ml stock. Final concentrations of AmB were higher for yeast peptone medium (range 125–500 ng/ml) than synthetic defined medium (range 63–250 ng/ml), with BY4741-derived strains sensitive at the top of these ranges and RS453–derived strains at lower concentrations. Growth was for 48–72 hr at 30°C.

## Sterol transport assays

Procedures for hypoxic incubation of yeast cells, quantification of DHE, DHE esters and ergosterol by reversed phase HPLC, quantification of radiolabeled cholesterol and cholesteryl esters by TLC, and sucrose gradient fractionation to measure transport of ergosterol from the ER to the PM were carried out as described previously (*Georgiev et al., 2011*) with the following modifications. For assays of the retrograde transport of DHE, cells were inoculated at $OD_{600} = 0.005$ and incubated in a hypoxic chamber for 36 hr prior to initiating the aerobic chase. For anterograde transport of [$^3$H]-ergosterol, the breaking buffer (*Georgiev et al., 2011*) additionally contained 10% (wt/vol) sucrose.

## Other assays

ACAT assays and isolation of detergent–insoluble complexes was carried out as described previously (*Georgiev et al., 2011*).

## Acknowledgements

We thank Will Prinz, Sean Munro for plasmids, Chris Stefan, Maya Schuldiner, Susan Ferro-Novick for strains, Sam Canis for support, and Catherine Rabouille and Sean Munro for comments on the manuscript.

## Additional information

### Funding

| Funder | Grant reference | Author |
| --- | --- | --- |
| Qatar National Research Fund (QNRF) | NPRP 5-669-1-112 | Yves Y Sere, Diana M Calderón-Noreña |
| Medical Research Council (MRC) | MR/J006580/1 | Louise H Wong |
| European Commission | Sphingonet ITN, project ref 289278 | Alberto T Gatta |

The funders had no role in study design, data collection and interpretation, or the decision to submit the work for publication.

## Author contributions
ATG, LHW, YYS, SC, TPL, Conception and design, Acquisition of data, Analysis and interpretation of data, Drafting or revising the article, Contributed unpublished essential data or reagents; DMC-N, Conception and design, Acquisition of data, Analysis and interpretation of data; AKM, Conception and design, Analysis and interpretation of data, Drafting or revising the article, Contributed unpublished essential data or reagents

## Author ORCIDs
Alberto T Gatta, http://orcid.org/0000-0002-2404-7351
Anant K Menon, http://orcid.org/0000-0001-6924-2698

# Additional files

## Major dataset
The following previously published dataset was used:

| Author(s) | Year | Dataset title | Dataset ID and/or URL | Database, license, and accessibility information |
| --- | --- | --- | --- | --- |
| Maureen E Hillenmeyer, Fung E, Wildenhain J, Sarah E Pierce, Hoon S, Lee W, Proctor M, Robert P St Onge, Tyers M, Koller D, Russ B Altman, Ronald W Davis, Nislow C, Giaever G | 2008 | Yeast Fitness Database (FitDB) | chemogenomics.med.utoronto.ca/fitdb/fitdb2.cgi | Publicly available at Yeast Fitness Database. |

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
