## [Decision Letter]

Thank you for sending your work entitled “A new family of StART domain proteins at contact sites has a role in ER-PM sterol transport” for consideration at *eLife*. Your article has been favorably evaluated by Randy Schekman (Senior editor) and three reviewers, one of whom is a member of our Board of Reviewing Editors.

The Reviewing editor and the other reviewers discussed their comments before we reached this decision, and the Reviewing editor has assembled the following comments to help you prepare a revised submission.

1) There is concern regarding the conclusion that Lam proteins mediate sterol exchange between the plasma membrane and endoplasmic reticulum. The reviewers appreciate that this is unlikely to be resolved in a single study, however, they feel that more definitive evidence that sterol is a physiological ligand for Lam proteins needs to be provided. Specific experiments to address this more rigorously are the following:

A) In panel 2C, the authors indicate that Lam4S2 shows saturable binding of DHE at increasing concentrations of liposomes that contain 30 mol% DHE. This is not obvious from this figure (no FRET values below 1 µM DHE, which is similar 2-6 µM); the experiments should be repeated with liposomes containing lower mol% of DHE in order to measure binding or use a fixed liposome concentration but with increasing mol% of DHE.

B) For the competition experiments presented in Figure 2, competing lipids and sterols are dispersed in methanol in aqueous buffer, which is not an appropriate experimental approach since the lipids will not be dispersed homogenously (likely just insoluble aggregates of unknown structure) and it is unclear how much DHE binding occurs under these conditions. Competing ligands should be presented in liposomes with a defined composition of DHE.

2) AmB sensitivity of *Lam* mutant strains is inconsistent with the observation that sterol levels do not change in the plasma membranes of mutant cells and, hence, are not fully consistent with the proposed role of Lam proteins in sterol transport. Again, the reviewers appreciate that this is unlikely to be resolved here, so they suggest that you present the lipid analyses of plasma membrane fractions, which are stated, but the data are not shown, in the manuscript. Related, the text should emphasize that it is presently unclear if the Lam proteins regulate or themselves mediate sterol levels in organelle membranes.

The reviewers also make several other suggestions that I encourage you to take to heart when revising your manuscript. In particular, regarding information to include (labels) in Figure 2, and the Discussion section, which they found to be confusing.

---

## [Author Response]

*1) There is concern regarding the conclusion that Lam proteins mediate sterol exchange between the plasma membrane and endoplasmic reticulum. The reviewers appreciate that this is unlikely to be resolved in a single study, however, they feel that more definitive evidence that sterol is a physiological ligand for Lam proteins needs to be provided. Specific experiments to address this more rigorously are the following*:

*A) In panel 2C, the authors indicate that Lam4S2 shows saturable binding of DHE at increasing concentrations of liposomes that contain 30 mol% DHE. This is not obvious from this figure (no FRET values below 1 µM DHE, which is similar 2-6 µM); the experiments should be repeated with liposomes containing lower mol% of DHE in order to measure binding or use a fixed liposome concentration but with increasing mol% of DHE*.

We see the validity of this point, and we have repeated the binding curve for Lam4S2 including concentrations of DHE liposomes down to 0.3x the lowest concentration used previously. Compared to the previous estimate of Kd as 0.61 µM (SD ±0.25), we have now obtained a value of 0.53 µM (±0.11), where the increased accuracy derives from the inclusion of the additional concentrations. This experiment is now shown in panel 2C.

*B) For the competition experiments presented in*
Figure 2*, competing lipids and sterols are dispersed in methanol in aqueous buffer, which is not an appropriate experimental approach since the lipids will not be dispersed homogenously (likely just insoluble aggregates of unknown structure) and it is unclear how much DHE binding occurs under these conditions. Competing ligands should be presented in liposomes with a defined composition of DHE*.

There may well be a problem of phospholipids not competing ideally with sterols when together added in methanol, although we note that this can only be quantitative, not qualitative, since PLs bound well to E-Syt2 (in preference to sterol) using precisely this approach (Schauder et al., Nature 2014). For the major phospholipids (PC, PS, PI and PE) we note that there was no extraction of radiolabelled lipid by Lam4S2 in Figure 2. Here all of PC, PS, PI and PE were present and labelled to levels equal to (or above) that of cholesterol. For the revised text we have analyzed the phosphorimager file, finding that these lipids were virtually absent from the Lam4S2 lane (relative recovery ≤0.3% compared to cholesterol). This places a lower limit on the Kd of binding of these lipids of approximately 100 µM.

We consider that the problem of dispersal after addition in methanol does not apply when comparing one sterol with another, so the sterol competition data from the original Figure 2 have been retained.

*2) AmB sensitivity of Lam mutant strains is inconsistent with the observation that sterol levels do not change in the plasma membranes of mutant cells and, hence, are not fully consistent with the proposed role of Lam proteins in sterol transport. Again, the reviewers appreciate that this is unlikely to be resolved here, so they suggest that you present the lipid analyses of plasma membrane fractions, which are stated, but the data are not shown, in the manuscript. Related, the text should emphasize that it is presently unclear if the Lam proteins regulate or themselves mediate sterol levels in organelle membranes*.

We agree with the points raised by the reviewers, and we have followed their suggestions.

We have included a new supplement for Figure 6 with 3 panels (Figure 6—figure supplement 1):

A: ACAT activity in strains lacking Ysp1p, Ysp2p or Sip3p.

B: Overall amount of ergosterol in PM fractions enriched in these single delete strains.

C: Sterol partitioning into detergent-insoluble complexes in the same strains, an empirical indicator of lipid organization.

All three panels confirm the results stated (but not shown) in the first draft.

The final section of the Discussion has been re-written completely. It emphasizes the lack of mechanistic certainty, beginning: “An important point to make is that we have yet to establish whether the StART-like proteins Ysp1p, Ysp2p and Sip3p regulate changes in sterol levels in organelle membranes, with an indirect role on traffic, or if they themselves directly mediate sterol transfer.”

We have included a short section in the Discussion (new paragraph five) that addresses the lack of correlation between sterol levels/extractability mentioned by the reviewers. This includes a brief description of the genetics of AmB sensitivity, in relation to which we have carried out one additional experiment, which shows *∆ysp1*/*∆ysp2*/*∆sip3* strains are sensitive to natamycin, a third polyene (result described, not shown, in relation to Figure 4 in the subsection headed “Ysp1p, Ysp2p, Sip3p and Lam4p target punctate ER-PM membrane contacts”).